# Assessment of the Reprocessed Suomi NPP VIIRS Enterprise Cloud Mask Product

Lin Lin [1,*] , Xianjun Hao [2] , Bin Zhang [1] , Cheng-Zhi Zou [3] and Changyong Cao [3]

1 Earth System Science Interdisciplinary Center, University of Maryland, College Park, MD 20740, USA; bzhangys@umd.edu
2 Environmental Science and Technology Center, Department of Geography and GeoInformatics, George Mason University, Fairfax, VA 22030, USA; xhao1@gmu.edu
3 Center for Satellite Applications and Research, NOAA/NESDIS, College Park, MD 20740, USA; cheng-zhi.zou@noaa.gov (C.-Z.Z.); changyong.cao@noaa.gov (C.C.)
* Correspondence: linlin@umd.edu

**Abstract:** The Visible Infrared Imaging Radiometer Suite (VIIRS) onboard the Suomi National Polar-orbiting Partnership (S-NPP) satellite continually provides global observations used to retrieve over 20 VIIRS Environmental Data Record (EDR) products. Among them, the cloud mask product is essential for many other VIIRS EDR products such as aerosols, ocean color, and active fire. The reprocessed S-NPP VIIRS Sensor Data Record (SDR) data produced by NOAA/Center for Satellite Applications and Research (STAR) have shown improved stability and consistency. Recently, the VIIRS Enterprise Cloud Mask (ECM) has been reprocessed using the reprocessed VIIRS SDR data. This study assesses the reprocessed ECM product by comparing the reprocessed cloud mask types and cloud probability with those from the operational VIIRS ECM product. It found that the overall differences are small. Most of the discrepancies occur between neighboring types at the cloud edge. These findings help lay the foundation for the user community to understand the reprocessed ECM product. In addition, due to the better quality of the reprocessed VIIRS SDR data that are utilized to generate the reprocessed ECM product, it is expected that the reprocessed ECM product will have better stability and consistency compared to the operational ECM products. Therefore, the reprocessed ECM product is a useful benchmark for the user community.

**Keywords:** SNPP VIIRS cloud mask reprocessing; assessment of reprocessed cloud mask types and cloud probabilities





## 1. Introduction

The Visible and Infrared Imaging Radiometer Suite (VIIRS) onboard the Suomi National Polar-orbiting Partnership (S-NPP) satellite was successfully launched on 28 October 2011. VIIRS has 22 spectral bands, covering the wavelengths from 0.4 to 11.8 μm and including 14 reflective solar bands (RSBs), seven thermal emissive bands (TEBs), and one day–night band (DNB) [1,2]. VIIRS level-1 Sensor Data Record (SDR) data are used as inputs to retrieve over 20 VIIRS Environmental Data Record (EDR) products, including snow/ice cover, clouds, fog, aerosols, fire, smoke plumes, vegetation health, sea/land surface temperature, ocean color, etc. All are required for environmental hazard monitoring and are useful in crucial economic sectors, such as transportation, fishing, energy, and agriculture, all of which impact human health [3].

VIIRS is performing very well on orbit since the Suomi NPP was launched. However, there have been many updates in the SDR calibration parameters and algorithms over the mission life [3–9]. The SDR products were in various levels of maturity during these years as the SDR data were declared at the beta, provisional, and validated maturity levels in April and October 2013, and March 2014, respectively. While long-term time series analysis requires consistent calibrated VIIRS data records, the historical calibration changes lead

to inconsistencies in the NOAA operational S-NPP VIIRS SDRs, which is not suitable for climate change monitoring and detection. In order to make VIIRS observations more usable for the science community, especially in climate study, S-NPP VIIRS SDRs from January 2012 to March 2020 have been reprocessed at NOAA/Center for Satellite Applications and Research (STAR) using the latest calibration algorithms [5,6,8–12]. The reprocessed VIIRS RSB SDR data removed the jumps and artificial oscillations caused by frequent changes in the operational calibration algorithms, especially in the early days of the mission when the algorithms were not mature [9,13,14]. The calibration is made consistent for the RSBs because: (a) the solar irradiance model has been updated (the Thuillier 2003 solar irradiance model [15] replaced the outdated model from Modtran previously); and (b) it reconciled the long-term degradation trends based on several independent calibrations including lunar calibration, Simultaneous Nadir Overpass (SNO) with MODIS, deep convective clouds (DCC), and vicarious site series comparisons using the Kalman filtering model that improves the time series consistency. For RSB M5 and M7, which are important to the clouds and aerosols, there was a 1.5% and 2.0% bias in the operational VIIRS SDR based on several studies. This has been corrected in the reprocessed VIIRS SDR. After reprocessing, the bias related to warm-up cool-down events is removed [6,8,13,14], the absolute accuracy for low light radiance has been largely improved for the DNB [10,13,14], the geolocation accuracy with 3-sigma uncertainty is better than 200 m, and terrain correction for VIIRS DNB was implemented to the days before 22 May 2015 [12]. A comprehensive summary of the VIIRS reprocessing and its preliminary validation results are provided in [14]. The reprocessed dataset is now available online from 2 January 2012 to 28 February 2017 (ftp: //jlrdata.umd.edu/pub/SNPP_Reprocessing/SDR/VIIRS/Baseline) and will be archived in the NOAA Comprehensive Large Array-data Stewardship System (CLASS) for end-users to access [14].

The cloud mask (CM) is a fundamental cloud property that is essentially needed in producing other EDR data products, such as aerosol properties, land surface reflectance, land surface temperature, sea surface temperature (SST), normalized difference vegetation index (NDVI), etc. Many cloud detection schemes using satellite observations have been proposed in the past [16]. Some of them are focused on specific regions [17,18], some are associated with specific EDR products, such as SST and vegetation properties [19,20], and some are suitable for global applications and general purposes [21–23]. NOAA Enterprise Cloud Mask (ECM) utilizes the Naïve Bayesian idea of clear/cloudy pixel detection that is applied to combined solar reflective and infrared imager and sounder data from a polar-orbiting satellite [24–26]. The high-resolution VIIRS cloud product (750 m) can help resolve the small convective elements that are sub-pixel for the MODIS cloud products (1km resolution) [27]. Moreover, MODIS will inevitably be replaced by VIIRS as it is getting close to the end of its life.

Many studies have been carried out to assess and improve the existing cloud detection algorithms. Okada et al. [28] and Nordkvist et al. [29] analyzed the cloud masking of SeaWiFS in coastal waters. Haglolle et al. [30] tested a multi-temporal cloud detection (MTCD) method for FORMOSAT-2 and LANDSAT, and proposed this method to be used for SENTINEL-2 level 2 processing. Andrew and Melin [31] evaluated the performances of different cloud mask schemes for the processing of NASA global ocean color data, and found that, for limited extreme conditions, a combination of different cloud masks is needed. Vermote et al. [32] analyzed the performance of the VIIRS cloud mask in surface reflectance (SR). Franch et al. [33] analyzed the improved AVHRR surface reflectance/NDVI version 4 product from the cloud mask improvements. Coluzzi et al. [34] analyzed the Sentinel-2 L1C cloud mask products under different biogeographic and cloudiness conditions. Frey et al. [35] developed the Continuity MODIS-VIIRS Cloud Mask (MVCM) algorithm to facilitate the continuity in the cloud detection between MODIS and VIIRS. Chi and Zhang [36] proposed an improved VIIRS dynamic threshold cloud detection algorithm (I-DTCDA) and compared its performance with that from the universal dynamic threshold cloud detection algorithm (UDTCDA) and the VIIRS cloud mask.

Recently, deep learning techniques have been applied for cloud detection. Chen et al. [37] developed a neural network cloud mask algorithm based on radiative transfer simulations and the validation results from collocated MODIS and the Cloud-Aerosol Lidar with Orthogonal Polarization (CALIOP) data show better performances than the MODIS cloud mask (MOD35 C6) over snow-covered areas in the mid-latitudes. Charles et al. [38] used the CALIOP data that are collocated with VIIRS to train a neural network for VIIRS cloud detection, and their model performs better in comparison with the VIIRS ECM during the nighttime and in high latitudes where the surface is covered by snow or ice. Sun et al. [39] built a data sample library from the observations of Airborne Visible Infrared Imaging Spectrometer (AVIRIS) with continuous bands between 400–2500 nm and used it to simulate different sensors based on the spectral response function, then employed the backpropagation (BP) neural network for cloud detection. Their results for VIIRS cloud detection have an overall accuracy of greater than 90%.

The current operational VIIRS ECM algorithm has been well validated. Kopp et al. [40] compared the first-year VIIRS cloud mask with CALIPSO and concluded that it satisfied the accuracy requirements of 2%. Heidinger [24] found that the VIIRS ECM has a probability of detection of 94%, a probability of missed clouds of 1%, a and probability of false cloud of 4% from the collocated CALIPSO data in 2012 and 2013. Comparison with the collocated MODIS cloud mask product (MOD35/MYD35) also shows that the ECM is performing well [24]. This study aims to evaluate the reprocessed VIIRS ECM products generated from using the reprocessed S-NPP VIIRS SDR data at NOAA/STAR. It's expected that the reprocessed VIIRS SDR will affect the ECM products because the VIIRS ECM algorithm employs observations of the VIIRS bands M5, M7, and M9–M16 and DNB observations (Table 1) [24,25].

**Table 1.** VIIRS, ABI and SEVIRI bands used in the ECM algorithm.

| VIIRS Band and Wavelength (μm) | ABI Band and Wavelength (μm) | SEVIRI Band and Wavelength (μm) |
|---|---|---|
| M5 (0.672) | 2 (0.64) | 1 (0.64) |
| M7 (0.865) | | |
| M9 (1.378) | 4 (1.37) | |
| M10 (1.61) | 5 (1.61) | 3 (1.64) |
| M12 (3.70) | 7 (3.90) | 4 (3.92) |
| | 9 (6.93) | |
| | 10 (7.34) | |
| M14 (8.55) | 11 (8.44) | 7 (8.70) |
| M15 (10.763) | 14 (11.21) | 9 (10.80) |
| M16 (12.013) | 15 (12.29) | 10 (12.00) |
| DNB (0.7) | | |

Reprocessing of VIIRS SDR has caused significant radiance changes for some bands. The operational M5 and M7 have been biased compared to Aqua MODIS similar channel observations. This bias has been corrected in the reprocessed SDR datasets with a reduction of 1.5% and 2% for M5 and M7, respectively. Meanwhile, changing the solar irradiance model in the reprocessed VIIRS SDR leads to a change in the radiance up to ~3% for some of the RSBs [13]. Additionally, seasonal cycles in the degradation of the solar diffusor have been corrected/smoothed in the reprocessed products compared to NOAA operational products. Furthermore, the infusion of different RSB calibrator results from the on-board solar diffusor, lunar view, deep convective cloud (DCC) histogram monitoring, and simultaneous nadir overpass (with MODIS) has been applied in the reprocessing RSB calibration. These changes in calibration coefficients and algorithms have helped to generate consistent,

improved radiance time series. How these changes in radiance values affect the cloud mask products needs to be evaluated for further applications of the reprocessed ECM in supporting generations of other EDRs.

The purpose of this paper is to compare the reprocessed S-NPP VIIRS ECM with the operational one and analyze the differences between these two ECM products. The paper is arranged as follows. Section 2 introduces the methodology for VIIRS ECM reprocessing. Data used in this study are also introduced in Section 2. Comparisons of the cloud mask types and cloud probability between the operational and reprocessed ECM products are discussed in Section 3. Section 4 is the summary and conclusions.

## 2. Method and Data

The S-NPP VIIRS ECM algorithm adopts some tests and thresholds from several other cloud mask algorithms, including those from the MOD/MYD35 MODIS cloud mask from University of Wisconsin-Cooperative Institute for Meteorological Satellite Studies (UW-CIMSS) [41], the Clouds and the Earth's Radiant Energy System (CERES) MODIS cloud mask from NASA Langley Research Center [42], the Cloud and Surface Parameter Retrieval (CASPR) cloud mask used in the AVHRR Polar Pathfinder Extended (APP-x) [43], and the GOES-R Baseline Cloud Mask [25].

To identify the cloud pixel that exhibits different characteristics from the clear-sky condition, VIIRS ECM employs 10 spectral and spatial tests at the pixel level, including Emissivity Referenced to the Tropopause (ETROP), the 11 μm Thermal Uniformity Test (BT11STD), the 11 and 12 μm Split-Window Test (BTD11_12), the Daytime 4 and 11 μm Thermal Contrast Test (BTD4_11_Day), the Nighttime 4 and 11 μm Thermal Contrast Test (BTD4_11_Night), 0.63 μm Reflectance (Ref0.63), the Relative Visible Contrast Test (RVCT), the Reflectance Ratio Test (Ref_ratio), the 1.38 μm Reflectance Test (Ref1.38), and the Normalized Difference Snow Index (NDSI) test [26]. Figure 1 shows the flowchart of the ECM algorithm. Each test produces a cloud or no cloud score, which is then used to determine whether a pixel is cloudy or clear by constructing the ECM using the Naïve Bayesian approach [24,26], which has been applied successfully to many complex detection problems [44]. Table 1 lists the VIIRS, ABI, and SEVIRI bands used in the ECM algorithm.

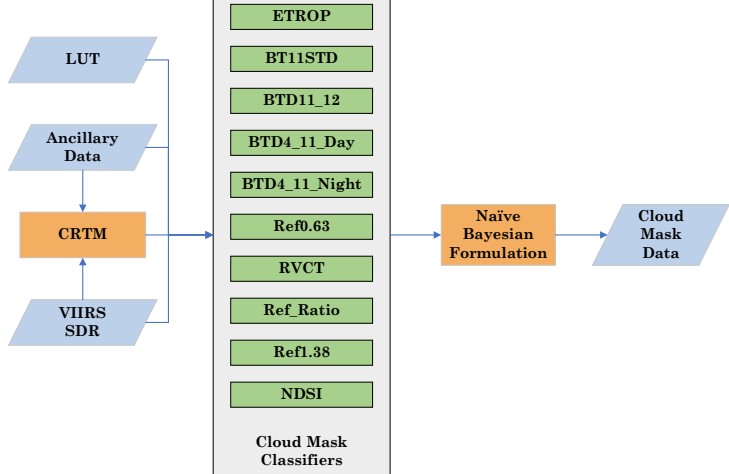

**Figure 1.** Flowchart of the ECM algorithm [26].

We used the STAR reprocessed VIIRS SDR [13,14] to generate the ECM products. Generating VIIRS ECM requires the sensor data of the calibrated solar reflectance percent (0–100%) for 0.63, 1.38, 1.64, 2.13 μm channels; longitude, latitude, calibrated radiances for 3.75 and 11.0 μm channels; calibrated lunar reflectance percent (0–100%) of the VIIRS; the derived 3.75 μm channel emissivity; 3.75 μm channel solar energy; sensor viewing zenith angle; solar zenith angle; relative azimuth angle; glint zenith angle; scattering angle; and solar zenith angles. All the above primary sensor data were obtained from the reprocessed

VIIRS SDR data. The generation of ECM also requires ancillary data, including surface type, surface elevation, land/coast/snow mask, ocean glint mask, numerical weather prediction (NWP) data, and Community Radiative Transfer Model (CRTM) simulations [24–26]. Consistent with the current operational ECM, we used the Optimum Interpolation Sea Surface Temperature (OISST, [45]), the output from the Global Forecast System (GFS) at 0.5-degree spatial resolution, CRTM 2.3, VIIRS Surface Type, and daily Interactive Multisensor Snow and Ice Mapping System (IMS)/SSMI snow map [46]. In addition, the operational ECM product was obtained from the NOAA/CLASS for the comparison tests in this study.

## 3. Results and Discussion

By applying the method with the ancillary and auxiliary data as described in Section 2 and running the operational ECM package using the S-NPP VIIRS reprocessed SDR data as input, we have produced the reprocessed ECM for the period from 1 April 2018 to 11 March 2020. The retrieved cloud properties in the ECM products include four cloud mask types (clear, probably clear, probably cloudy, cloudy), and cloud probability (0.0–1.0). In this preliminary study, we investigated the differences between the operational and reprocessed ECM products for the selected day on 1 May 2018 and chose six granules for detailed comparisons in the cloud mask types and cloud probability. These six granules are located close to the southeast corner of Australia and were chosen under the considerations to (1) avoid the Antarctic region (60S~90S), because the accuracy of ECM in high latitudes (>60°) is less than that between 60S and 60N [26], and (2) include both land and ocean surfaces. There are 14,745,600 pixels included in the six granules on 1 May 2018.

### 3.1. Cloud Mask Type Comparison

Figure 2 shows the cloud types from the operational (Figure 2a) and reprocessed ECM products (Figure 2b) separately, the pixels that have different cloud mask types (Figure 2c), and the concurrent VIIRS true-color image (Figure 2d). The mismatched pixels are color-coded in Figure 2c, based on the cloud types of the operational and reprocessed ECM product for each pixel, and the number of each scenario is listed in Table 2. It is found that 99.77% (14,711,249 out of 14,745,600 pixels) of the cloud mask types are the same; 0.23% (34,351 out of 14,745,600 pixels) are mismatched. Additionally, 99.97% (34,342 out of 34,351 pixels) of the mismatches occur between two neighboring types, i.e., "clear" and "probably clear", "probably clear" and "probably cloudy", "probably cloudy" and "cloudy". It is also noticed from the zoomed-in plot (Figure 2e) that the mismatches of cloud mask types mostly occur at the edge of the clouds (Figure 2d), and the zoomed-in plot (Figure 2e) illustrates it more clearly.

For the whole day of 1 May (Table 3), consistent results are obtained. 99.776% (2,481,513,523 out of the total 2,487,091,200 pixels) of the cloud mask types are the same, and 0.224% (5,577,677 out of the total 2,487,091,200 pixels) are mismatched. In addition, 98.67% (5,505,148 out of 5,577,677) of the mismatches occur between two neighboring types.

**Table 2.** The number of pixels in different cloudy mask types for six granules of the operational and reprocessed ECM products on 1 May 2018 (Total number of pixels: 14,745,600). The colors indicate the different scenarios with the cloud mask type of the operational (in row) and reprocessed (in column) ECM products.

| Operational / Reprocessed | Clear | Probably Clear | Probably Cloudy | Cloudy |
|---|---|---|---|---|
| **Clear** | 4,227,726 | 10,150 | 0 | 0 |
| **Probably Clear** | 1935 | 380,084 | 9156 | 0 |
| **Probably Cloudy** | 3 | 2049 | 515,177 | 8617 |
| **Cloudy** | 4 | 2 | 2435 | 9,588,262 |

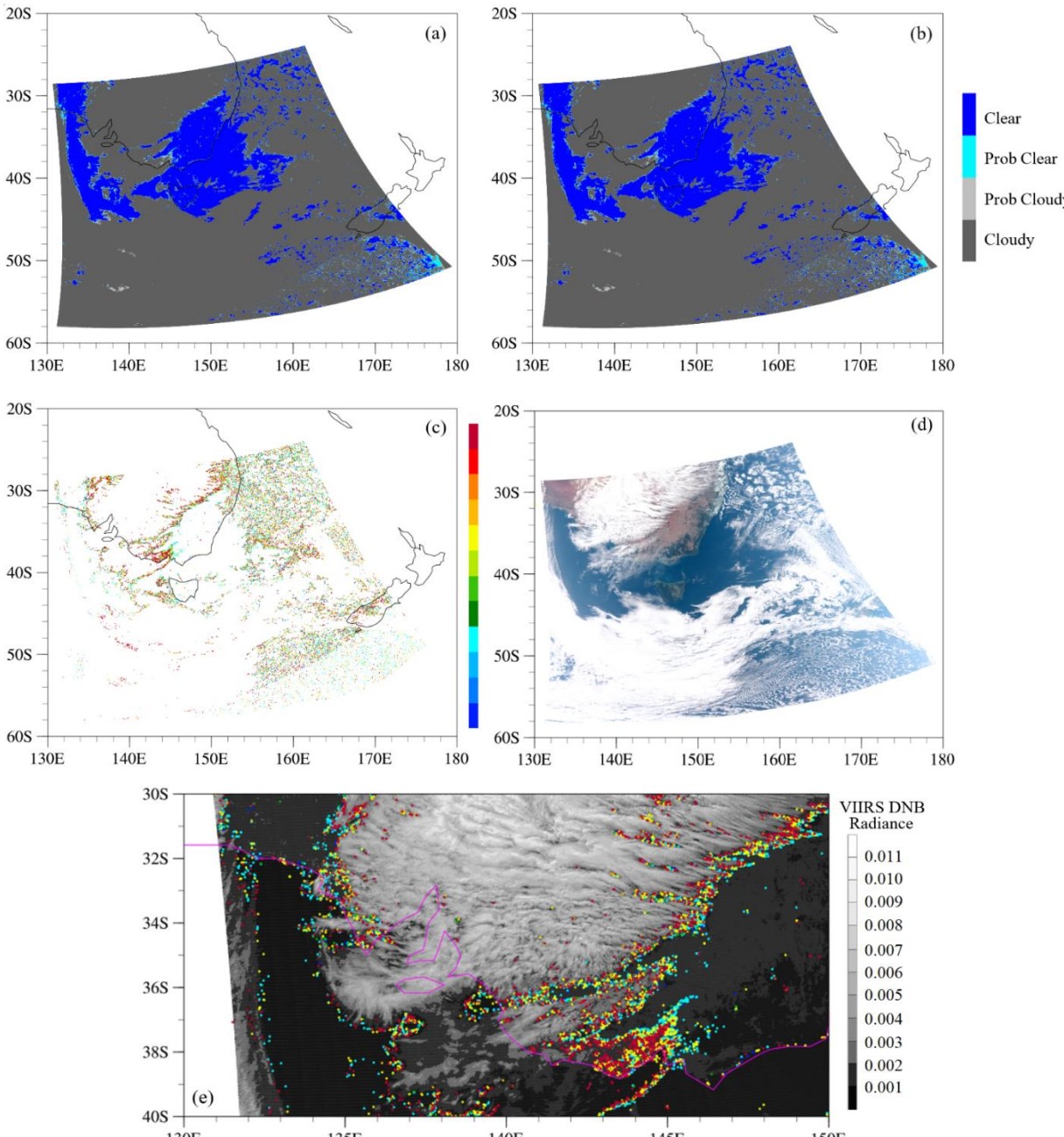

**Figure 2.** Cloud mask types of (**a**) the operational and (**b**) the reprocessed ECM products for six granules on 1 May 2018. (**c**): pixels that have different cloud mask types between the operational and reprocessed ECM product. (**d**): Concurrent VIIRS true-color image. (**e**): A zoomed-in plot of the pixels that have different cloud mask types overlaid on the VIIRS DNB radiance (unit: W/cm$^2$·sr$^{-1}$) within the domain of (30S~40S, 130E~140E). The pixel is color-coded based on the cloud mask type of the operational and reprocessed ECM products for each pixel, which is indicated in Table 2.

**Table 3.** The number of pixels in different cloud mask types of the operational and reprocessed ECM product on 1 May 2018 (total number of pixels: 2,487,091,200).

| Operational / Reprocessed | Clear | Probably Clear | Probably Cloudy | Cloudy |
|---|---|---|---|---|
| **Clear** | 739,311,193 | 1,424,287 | 18,845 | 4808 |
| **Probably Clear** | 732,226 | 149,238,446 | 1,141,409 | 41,737 |
| **Probably Cloudy** | 5117 | 587,786 | 185,431,688 | 1,071,928 |
| **Cloudy** | 385 | 2022 | 547,127 | 1,407,532,196 |

### 3.2. Cloud Probability Comparison

Figure 3 shows the cloud probability of the operational and reprocessed ECM products separately, which indicates different cloud probability in the pixels. Again, it is noticed that the mismatches of cloud probability mostly occur at the edge of clouds. Table 4 summarizes the number of pixels in Figure 3 based on its cloud probability of the operational and reprocessed ECM products. It is found that 92.58% (13,651,150 out of the total 14,745,600 pixels) of the cloud probability from these six granules are the same, and 7.42% (1,094,450 out of the total 14,745,600 pixels) are mismatched. The changes in the VIIRS SDR data lead to small differences in cloud probability, and 87.11% (12,844,713 out of 14,745,000 pixels) of the pixels having cloud probabilities of exactly 0 or 1 are the same. On the other hand, 57.58% (1,094,450 out of 1,900,887 pixels) of the pixels having cloud probabilities between 0 and 1 are different. Consistent results are obtained for the whole day of 1 May (figure omitted).

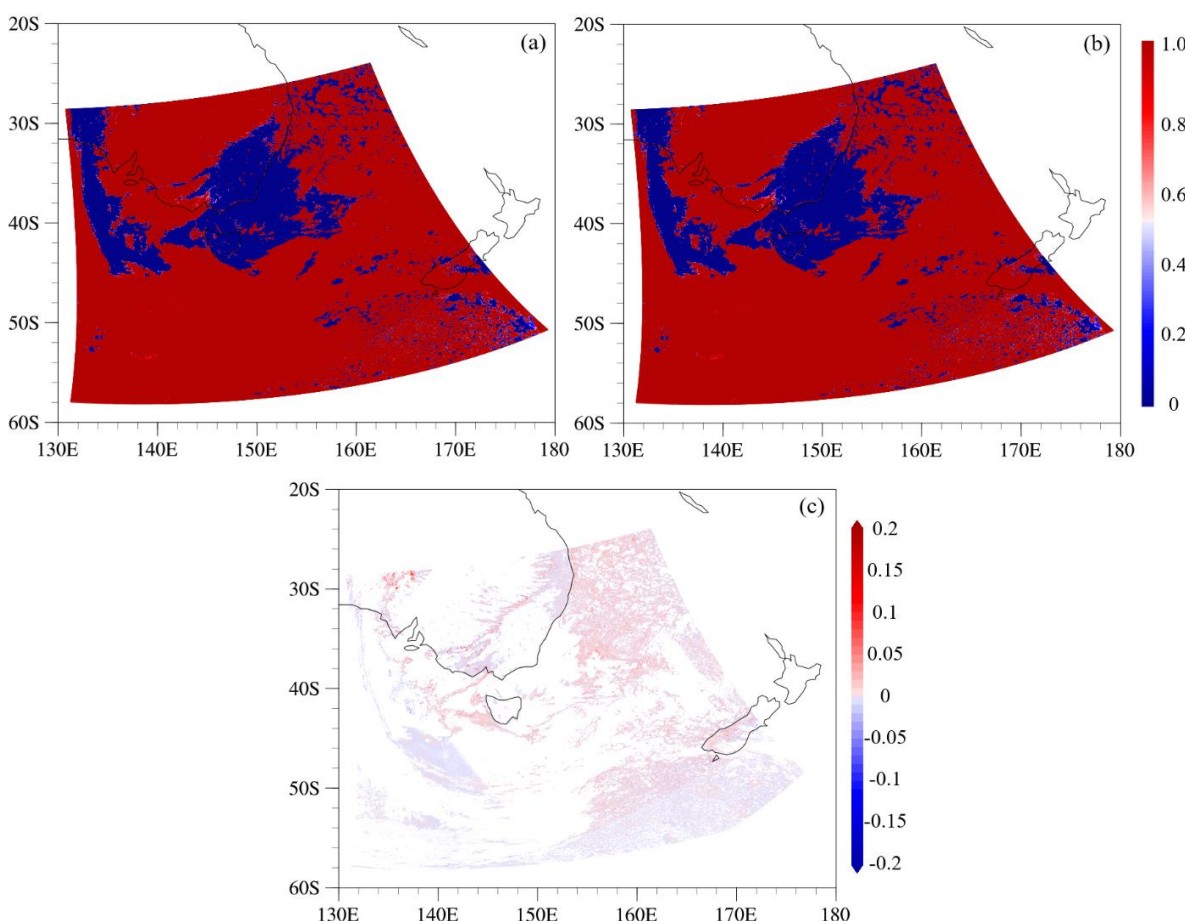

**Figure 3.** Cloud probability for (**a**) the operational and (**b**) the reprocessed ECM products for six granules on 1 May 2018. (**c**): pixels that have different cloud probability between the operational and reprocessed ECM products. The pixel is color-coded based on the difference between the probability of the reprocessed and operational ECM products.

**Table 4.** The number of pixels in different cloud probability for six granules of the operational and reprocessed ECM products on 1 May 2018 (total number of pixels: 14,745,600). The number in the bracket indicates the number of mismatches of cloud probability.

| Reprocessed | Operational | 0 | (0, 0.5] | (0.5, 1.0) | 1.0 |
|---|---|---|---|---|---|
| 0 | | 4,140,839 | 10,520 | 0 | 0 |
| (0, 0.5] | | 1944 | 466,592 (292,216) | 9156 | 0 |
| (0.5, 1.0) | | 0 | 2058 | 1,396,398 (764,337) | 10,843 |
| 1.0 | | 0 | 0 | 3376 | 8,703,874 |

Examination of the cloud probability of the reprocessed ECM product with respect to that of the operational ECM product for the mismatched pixels (Figure 4a) shows that, for the majority (67.3%) of the mismatched pixels, the cloud probability of the operational ECM product is larger than that of the reprocessed ECM product. In addition, 98.92% (1,082,674 out of 1,094,450) of the mismatches have a difference in cloud probability of less than 0.1, and 99.93% (1,093,666 out of 1,094,450) of the mismatches have a cloud probability difference less than 0.2 (Figure 4b).

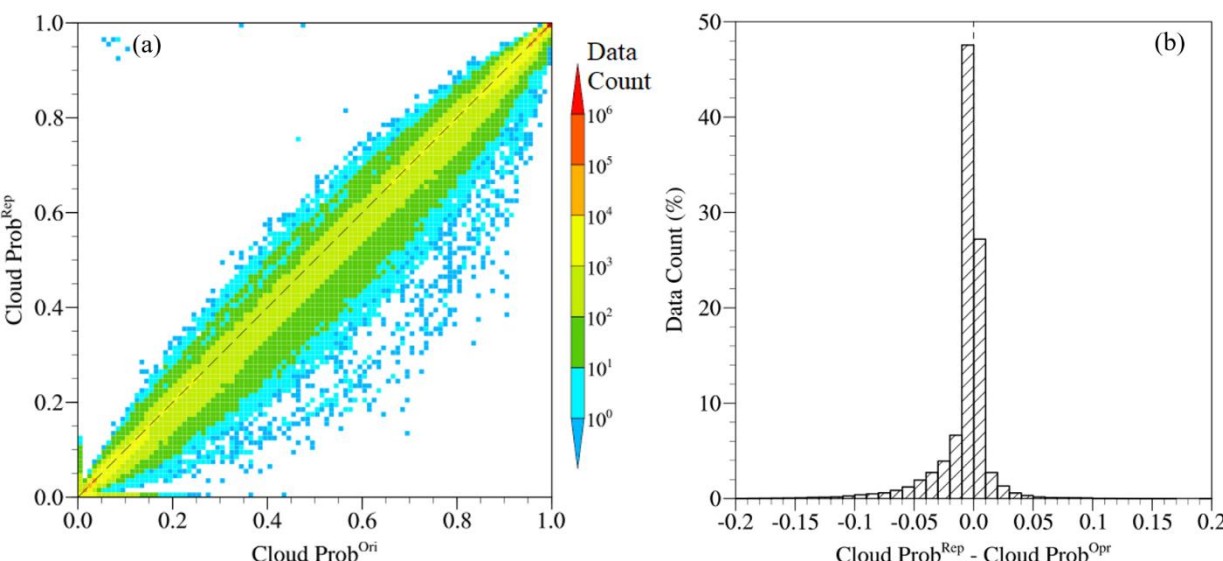

**Figure 4.** (**a**): Scatter plot of the cloud probability of the reprocessed ECM product with respect to the cloud probability of the operational ECM product, and (**b**): Frequency distribution of the difference between the cloud probability for the mismatched pixels of the reprocessed and operational ECM product for the six granules on 1 May 2018 (1,094,450 pixels in total).

## 4. Discussion and Conclusions

Using the newly reprocessed S-NPP VIIRS SDR data as inputs, we reprocessed the VIIRS ECM product on the current operational ECM software platform for the period of 1 April 2018 to 11 March 2020. This study assessed the cloud mask types and cloud probability of the reprocessed VIIRS ECM product by comparing them to the operational ECM product. It found that more than 99% of the pixels have the same cloud mask type, and more than 90% of the pixels have the same cloud probability. Most of the mismatches occur at the edge of clouds, and between two neighboring cloud mask types. The majority of the cloud probability mismatches are within "cloud" or "clear" cloud mask types. Additionally, about 99% of the mismatches have a cloud probability difference of less than 0.2. These findings lay a foundation for the user community to understand the reprocessed ECM

product. In addition, even though the difference to the operational ECM products seems to be small, the reprocessed ECM product is a useful benchmark for the user community.

We expect that the reprocessed ECM products will inherit stability and consistency in the reprocessed SDRs. It is expected that the time series of the cloud type for long-standing clouds (e.g., the Intertropical Convergence Zone, the polar stratus clouds) will be different between the operational and reprocessed ECM products. Future studies will be carried out to investigate this aspect using more data.

**Disclaimer**

The scientific results and conclusions, as well as any views or opinions expressed herein, are those of the authors and do not necessarily reflect those of NOAA or the Department of Commerce.

**Author Contributions:** Conceptualization, L.L. and C.-Z.Z.; methodology, L.L., C.-Z.Z., and C.C.; Software, L.L., X.H. and B.Z.; Data Curation, L.L., X.H. and B.Z.; Validation, L.L. and X.H.; Formal analysis, L.L.; Investigation, L.L.; Resources, C.-Z.Z. and C.C.; Writing—original draft preparation, L.L.; Writing—review and editing, X.H., B.Z., C.-Z.Z. and C.C.; Supervision, C.-Z.Z. and C.C.; Project Administration, C.-Z.Z. and C.C.; Funding acquisition, L.L. and C.-Z.Z. All authors have read and agreed to the published version of the manuscript.

**Funding:** This research was funded by NOAA (Grant #NA19NES4320002) to the Cooperative Institute for Satellite Earth System Studies (CISESS) at the Earth System Science Interdisciplinary Center (ESSIC), University of Maryland.

**Data Availability Statement:** The operational S-NPP VIIRS Cloud Mask data is publicly available via NOAA/CLASS (https://www.avl.class.noaa.gov/saa/products/welcome, accessed on 24 June 2021). The reprocessed S-NPP VIIRS SDR data are available at ftp://jlrdata.umd.edu/pub/SNPP_Reprocessing/SDR/VIIRS/Baseline/, accessed on 24 June 2021, and will be available at NOAA/CLASS next year. The reprocessed S-NPP VIIRS Cloud Mask data can be obtained from ftp://jlrdata.umd.edu/pub/SNPP_Reprocessing/EDR/Cloud_Mask/Baseline/, accessed on 24 June 2021.

**Acknowledgments:** The authors would like to acknowledge the technical support of the NOAA ASSIST team for setting up the ECM package.

**Conflicts of Interest:** The authors declare no conflict of interest.

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
