# Peer review of "Assessment of the Reprocessed Suomi NPP VIIRS Enterprise Cloud Mask Product"

_remotesensing, doi:10.3390/rs13132502_

Round 1

Reviewer 1 Report

I thank the authors for taking my initial review into consideration.  All of the changes to manuscript are satisfactory and I have no further comments.

Author Response

Thank you very much for your valuable comments! They helped us improve the manuscript greatly.

Reviewer 2 Report

As I noticed authors have added some extra literature in the introduction but in the conclusion they insist that “the reprocessed ECM product is highly recommended to the user community”. This is a statement which, according to my opinion, is not justified by the results they present.

In addition, the statement that the reprocessed ECM product presents “better stability and consistency of the reprocessed VIIRS SDR data that are utilized to generate the reprocessed ECM product” is not obvious to the reader by the analysis they perform.

Due to these statements-conclusions I believe that the manuscript should be rejected, although the assessment they have performed would be very useful to the user community.

Reviewer 3 Report

The authors have adequately addressed the comments raised; the manuscript has been improved. It is ready to be published.

Author Response

(The authors gave the same response as above.)

Round 2

Reviewer 2 Report

No further corrections are needed. Please accept the manuscript as it is.

This manuscript is a resubmission of an earlier submission. The following is a list of the peer review reports and author responses from that submission.

Round 1

Reviewer 1 Report

In this work, the authors perform an assessment of the differences in the newly reprocessed VIIRS Enterprise Cloud Mask (ECM) product with the current operational product.  The reprocessed SDR data provides a number of calibration improvements which is said will transfer onto the ECM product.  The authors briefly described the ECM algorithm and showed a comparison of select data.  It was found that the changes in the ECM were small, with most of the changes occurring for “neighboring” cloud types.  This assessment is important for the user community and will be a valuable addition to the literature.  While the results are presented satisfactorily and show the differences due to the product update for the selected scenes, some more background details and clarifications are needed for this work.  Below is a list of my specific comments.

- Line 50 - It would be best to be more detailed about what “better consistency” entails.  Was this assessed by looking at variation over selected sites, long-term drift, or bias?  Perhaps a combination of all three.  A one sentence description should be sufficient.  I will also point out here that the sentence refers to an enhancement of the RSB consistency but the citation refers primarily to TEB calibration improvements.  Is this a mistake in the reference or is RSB discussed significantly in this work?

- Lines 68-70 - The resolution of the MODIS and VIIRS products here should be listed for comparison.  The bands mentioned for VIIRS are all 750-m resolution and MODIS typically uses 1-km bands for these tests, which would not be a significant resolution change.  However, I am aware of some cloud mask products with 5+ km resolution, so maybe it is comparing to those.  In any case, clarification is needed here.

- Line 75 - Some example should be given for what “significant radiance changes” means.  This could be something like “up to X% for band MX”.  Is this in variation, drift, or bias?  Also, when discussing the specific tests that are performed below, it should be mentioned which tests are expected to be the most impacted by the changes in the SDR data.

- Section 2 - Were any of the thresholds for the algorithms used developed using VIIRS data?  If so, do these threshold values need to be re-evaluated in light of the reprocessed SDR data?

- Section 2 - A table of the relevant VIIRS bands with their wavelengths should be included.  This is because throughout section 2, all of the tests are listed by wavelength.  Also, since tests are adopted from other instruments as mentioned in the first paragraph of this section, it may be useful to list the equivalent bands from those instruments in this table as well.  Finally, if it makes sense to do so, a final column listing the tests where each band is used might also be helpful for reference.  However, this may become someone cluttered, so I’ll leave it to your judgement as to whether it makes sense to include this final column or not.

- Section 2, Final Paragraph - In order to better understand all of the inputs of the algorithm, a high level flow chart (no detail needed about the mechanisms of specific tests) would be very helpful showing the inputs, the number of tests performed and how they are used in the final scoring of the pixel cloud type/probability.

- Figure 1 - If this example granule is from daytime data, a 4th panel with a true-color image would be nice to see if possible.

- Figure 1 - The figures (a) and (b) are largely the same (which I know is the point), but it might be nice to have a zoomed-in inset figure for some region near a cloud edge where the differences can be seen more clearly.  There would be space in the corners of the figures to include this.

- Line 164 - Can you clarify what is meant by having 92.58% of the pixels have the same probability?  If the probabilities are floating point values, I would expect radiance changes to produce small differences in probability for most of the pixels.  However, if there is some threshold set on what is considered to be “the same”, it should be stated.  Also, it is possible that all of these pixels have probabilities of exactly 0 or 1, in which case small changes to the SDR might not result in changes to those numbers.  However, in between 0 and 1 I would expect changes to occur for most pixels.

- Figure 2 - Based on the above statement that 92.58% of the pixels have the same probability, subfigure (c) seems to indicate a much greater fractional difference for this granule just based on the area covered by the image.  It seems unlikely that this can be explained by pixel size differences based on the projection of the image.  However, it would be possible that this is an image rendering artifact and that in between points of changes there are many points that are the same that remain unseen in the plot.  Can you comment on this?

- Figure 3(a) - Although this image is already square, a thin 1:1 line would probably be beneficial to add to the plot.

- Line 195 - It is mentioned that the ECM should inherit the stability of the SDR reprocessed data.  In what way is the time series stability of the ECM assessed and what expected changes are there from using the new SDR data.  Its acceptable to not have this assessment completed, but it would be good to have some indication of what potential issues there are in the operational ECM product that are solved by using the reprocessed SDR data as an input.

Reviewer 2 Report

As a general comment I would like to declare that I am not convinced that the reprocessed ECM products offer significant improvements in relation to the operational data so I contradict authors’ conclusion that reprocessed ECM product must be highly recommended. The presented comparison of reprocessed and operational data shows no or very small differences and I suppose that this situations is the most common one.

Specific comments:

Abstract: it is missing the meaning of the abbreviation ECM

line 74: Give a table with details about VIIRS band otherwise remove bands’ names.

line 104: “contrast” instead of “contract”

Reviewer 3 Report

This manuscript aims to compare and discuss cloud mask types and cloud probability between the operational and reprocessed Enterprise Cloud Mask (ECM) products. It has a good quality presentation but is weak and poor in the content to be published in Remote Sensing journal in this form.

First of all, the theme is not original (for that I suggest to mention similar papers in the introduction) but I think that it could be of interest for part of the remote sensing community (such as Suomi NPP VIIRS Enterprise users).

Moreover, which is exactly the analyzed data set? What do you mean with “we have produced the reprocessed ECM for the period from April 1, 2018, to March 11, 2020. In this study, we investigated the differences and consistency between the operational and reprocessed ECM products for the selected day on May 1, 2018 and randomly chose six granules” (lines 129-132)? Please, deepen this description.

For example, you select random six granules: why this choice? Where are they localized? Do you  describe the results about only one day (May 1, 2018) or not? And why? This section is too much confused. Please, make more detailed descriptions.

Section 2 -Methodology: I suggest to insert a flowchart to better understand the method used to process data and please, specify which cloud mask you consider as true in your analysis.

Finally, authors often repeat that “the mismatches of cloud probability mostly occur at the edge of clouds”. Please, add a figure to show this result which seems to be the mainly important in this work.

Minor:

  • Abstract: please, define ECM
  • Line 119: Please, define NWP

Round 2

Reviewer 2 Report

Authors replied to my specific comments, however the other changes they have made in manuscript do not alter my previous opinion that the reprocessed ECM products offer significant improvements in relation to the operational data. To my opinion, manuscript does not present great interest for the readers so I think that it should be rejected. 

Reviewer 3 Report

I did not find answers about the following comments:

First of all, the theme is not original (for that I suggest to mention similar papers in the introduction) but I think that it could be of interest for part of the remote sensing community (such as Suomi NPP VIIRS Enterprise users)”.

Please, add more literature. In particular, I suggest again to add papers about cloud mask assessment such as, for example:

  • Lin Sun, Xu Yang, Shangfeng Jia, Chen Jia, Quan Wang, Xinyan Liu, Jing Wei & Xueying Zhou (2020) Satellite data cloud detection using deep learning supported by hyperspectral data, International Journal of Remote Sensing, 41:4, 1349-1371, DOI: 10.1080/01431161.2019.1667548
  • Coluzzi, R., Imbrenda, V., Lanfredi, M., & Simoniello, T. (2018). A first assessment of the Sentinel-2 Level 1-C cloud mask product to support informed surface analyses. Remote sensing of environment217, 426-443.
  • Andrew Clive Banks & Frédéric Mélin (2015) An assessment of cloud masking schemes for satellite ocean colour data of marine optical extremes, International Journal of Remote Sensing, 36:3, 797-821, DOI: 10.1080/01431161.2014.1001085
  • Evaluation of Visible Infrared Imaging Radiometer Suite (VIIRS) neural network cloud detection against current operational cloud masks. White, Charles H; Heidinger, Andrew K; Ackerman, Steven A. Atmospheric Measurement Techniques; Katlenburg-Lindau Vol. 14, Fasc. 5,  (2021): 3371-3394. DOI:10.5194/amt-14-3371-2021